# The Development and Application of Drama-Combined Nursing Educational Content for Cancer Care

**DOI:** 10.3390/ijerph18189891

**Published:** 2021-09-20

**Authors:** Eunyoung E. Suh, Jeonghee Ahn, Jiyoung Kang, Yoonhee Seok

**Affiliations:** 1Center for Human-Caring Nurse Leaders for the Future by Brain Korea 21 (BK 21) Four Project, Research Institute of Nursing Science, College of Nursing, Seoul National University, Seoul 03080, Korea; esuh@snu.ac.kr; 2Center for Human-Caring Nurse Leaders for the Future by Brain Korea 21 (BK 21) Four Project, College of Nursing, Seoul National University, Seoul 03080, Korea; ajh627@snu.ac.kr; 3Research Institute of Health and Nursing, College of Nursing, Jeju National University, Jeju 63243, Korea; jiyoungkang@jejunu.ac.kr

**Keywords:** nursing education, drama, teaching, nursing student

## Abstract

With the recent increase in the number of cancer patients, it is important to educate nursing students using pedagogical techniques that nurture understanding and empathy for cancer patients. This study examined nursing students’ experiences caring for cancer patients after receiving drama-combined nursing education for cancer care (DCC), which consisted of three elements: lectures, dramatic scenarios, and debriefing. The lectures dealt with cancer statistics, diseases, and nursing, and the dramatic scenarios depicted both breast cancer patients and lung cancer patients. Sixty-seven junior-year nursing students attended a 90 min DCC session developed by the authors. Focus group interviews were conducted to explore students’ educational experiences, and the following three themes were derived using the thematic analysis method: ‘understanding the lives of patients with severe diseases and their families’, ‘seeing a nursing role model provide patient-centered care’, and ‘projecting an image of oneself as a future nurse’. Using drama in nursing education for cancer patients provided an opportunity for students to imagine the clinical experiences of cancer patients, helping them to understand patients’ points of view and reflect on their self-images as future nurses. The DCC developed for nursing students in this study is a promising way to deliver distinctive and meaningful learning experiences.

## 1. Introduction

The National Cancer Institute estimated that the number of cancer survivors in the United States in 2019 was 16.9 million, and that number is expected to increase to 22.2 million by 2030 [1]. As a result of this increase in the number of cancer patients, nursing students are more likely to encounter cancer patients in clinical practice. Therefore, as preparation for their future careers, nursing students should thoroughly understand cancer care and be prepared to empathize with and care for cancer patients.

However, the growing emphasis on patient safety and rights has restricted opportunities for nursing students to directly perform nursing care for patients in clinical settings [2]. In particular, concerns about patient infection in the context of the COVID-19 pandemic have caused clinical practice to become more observation-oriented. These changes in the clinical practice environment have further reduced the number of opportunities for nursing students to interact directly with patients, providing an obstacle for students to obtain dynamic clinical experience in the field [3]. In addition, clinical practice experiences differ among students depending on timing and context [4]. The fragmentation and inconsistency of clinical practice experiences can make it difficult for nursing students to fully understand and empathize with cancer patients and restrict their opportunities to develop relationships with patients.

In order to address problems related to clinical practice, in the 1990s, nursing education programs introduced simulation-based education, which provides students with opportunities to practice patient care with simulated patients (also known as standardized patients). However, clinical practice using standardized patients in simulation-based education tends to focus on isolated basic nursing skills or simple practices, such as physical assessment, medication, basic life resuscitation, and airway management [5], making this framework insufficient for accumulating experience interacting with patients and developing empathy [6]. Beginning in the 2000s, attempts have been made to incorporate tablets, mobile devices, and virtual reality into nursing education [7], but barriers such as insufficient Wi-Fi access and device availability still exist. Moreover, teaching methods using mobile devices have limitations, including a lack of student motivation, insufficient interaction between learners and instructors, and an incomplete ability to portray the reality of the clinical setting [8]. Developing optimal nursing education by linking theory and practice therefore remains a challenge for instructors in charge of nursing education [9].

One way for students to build experience in an evidence-based clinical practice environment is to utilize drama, an artistic tool [3]. Education using drama is a teaching method directed toward students for the purpose of achieving educational goals, rather than performing in front of an audience. Dramatic elements can be used to educate viewers and participants about the lived experiences of patients. Through drama, nursing students can identify with characters in a performance, engage in reflective thinking to solve problems, and consider solutions to the problems portrayed in a performance [10]. Thus, the dramatic method of indirectly experiencing and exploring realistic situations in a virtual and safe environment can be a useful teaching method [11].

Education using drama for psychiatry students has been shown to convey knowledge of methods for effectively communicating with patients and patients’ families and to improve students’ confidence when communicating with patients [12]. The use of drama in nursing education provides students with opportunities for self-reflection related to nursing and improves students’ critical skills in clinical situations [13]. It was also found that, when using drama, nursing students were able to explore themselves professionally as nurses, understand patients’ points of view, and improve their ability to empathize with patients [14]. Education using drama helped students to achieve an understanding of theoretical concepts and approaches to real-world situations by dramatically reenacting end-of-life care scenarios and situations requiring conflict management using techniques that cannot be learned via theoretical education alone [15,16]. The safe clinical environment reproduced in dramatic scenarios can provide nursing students with a comfortable learning environment and opportunities to reflect, helping them accumulate new knowledge while switching between real-life and simulated situations [17]. In addition, education that incorporates drama can promote experiential learning and nurture creativity and critical thinking in students [3]. However, no previous research has yet investigated the use of drama in nursing education related to cancer care. Therefore, this study aimed to examine the experiences of nursing students after they took part in an educational program—hereafter referred to as drama-combined nursing education for cancer care (DCC)—that incorporated lectures and dramatic depictions of the entire process of cancer trajectory, including cancer diagnosis, treatment, symptom management, and end-of-life care.

## 2. Materials and Methods

### 2.1. Study Design and Approach

This is a qualitative study exploring the experiences of nursing students through focus group interviews (FGIs) after they took part in DCC education. FGIs are a method for comprehensively collecting participants’ opinions and conducting in-depth examinations of the similarities and differences in individuals’ experiences [18]. After voice-recorded interview data were transcribed, the researcher read the transcriptions several times and analyzed them using qualitative thematic analysis, which identifies main themes by examining relationships between concepts identified in the interview transcriptions, focusing on people’s views, opinions, knowledge, experiences, or values [19].

### 2.2. Participants

The participants of this study were 67 nursing students in their junior year at a nursing college in Seoul, South Korea, who took part in the DCC program. The participants had already received education on basic nursing skills, communication skills, nursing diagnosis, and practice with simulated patients at the time of participation in this study. After participation in the DCC program, 51 students ultimately agreed to share their thoughts in an FGI and participate in the study.

### 2.3. Development and Implementation of DCC

#### 2.3.1. Team Roles in Developing and Implementing DCC

DCC was developed by a nursing professor and three graduate students who decided on the structure of the course and designed its content. A professor in the Department of Theater and Film wrote scripted scenarios and directed the performance. Six professional stage actors participated in the performance: four actors played the roles of a patient, a guardian, a nurse, and a doctor, while the other two actors were in charge of assistant direction, stage setting, and props. The actors received education about the entire treatment cycle of cancer patients, the symptoms and prognosis of cancer treatment, and family dynamics arising from cancer treatment.

#### 2.3.2. Composition of the Performance

The DCC scenarios included breast cancer patients and lung cancer patients. Five scenes were performed for each of those two patient categories, and a brief lecture was conducted beforehand to inform students of the nursing-related concepts relevant to the performance. The fictional breast cancer patient was a woman in her mid-40s with a husband, a daughter in middle school, and a son in elementary school. The fictional lung cancer patient was a man in his late 50s with a wife and a daughter in high school. The scenarios portrayed five different points in the process of cancer treatment (diagnosis, treatment, survivor management, palliative care, and end-of-life care), so that students could see a patient’s entire disease process [20].

The first scene involving nursing care for the breast cancer patient portrayed the patient and her caregivers at a clinic receiving the news of her cancer diagnosis for the first time. The second scene showed the patient suffering from postoperative pain after cancer-related surgery and a caregiver taking care of the patient. The third scene depicted the patient suffering from chemotherapy-induced nausea and vomiting and demonstrated nursing interventions for chemotherapy side effects. The fourth scene depicted the patient and family coping with the news that the cancer had metastasized to the lungs. The fifth scene showed a therapeutic communication situation in the context of hospice and end-of-life care. The same disease process was depicted in the lung cancer scenario (Table 1).

#### 2.3.3. Implementation of the DCC Classes

The DCC class was provided to the students as a compulsory nursing subject. Students who took the practicum of adult health nursing automatically participated in the DCC. The DCC classes were held in a lecture hall at the nursing college, and the set was constructed in the front of the room (Figure 1). After the DCC, the students were asked if they would agree to participate in the focus group interviews. The class lasted 90 min in total and alternated between six 5-min lectures providing background information (comprising 30 min in total) and five dramatic scenes (50 min in total), followed by a debriefing and a Q&A (10 min) at the end. A total of 67 participants were divided into 6 groups, and the DCC classes were provided biweekly across a total of 6 sessions during the fall semester of 2015.

While the dramatic scenarios were being performed, the audience of nursing students participated as learners, watching and listening to the scenes and lectures. In each dramatic scene, the audience of students was exposed to the following: pain control, oxygen therapy, therapeutic communication, nursing for chemotherapy-induced nausea and vomiting, emotional support of the family, hospice care, and end-of-life care.

### 2.4. Data Collection

#### 2.4.1. Data Collection Procedure

In order to examine the nursing students’ experiences with DCC, five FGIs were conducted, with 9–12 students in each focus group. The focus group interviews were conducted right after each DCC class. FGIs were performed in a quiet and comfortable place, such as a seminar room for 1 h. The interviews were recorded with a digital recorder.

#### 2.4.2. Focus Group Interviews

All interviews were conducted by one nursing college professor and one PhD student. The FGIs lasted for one hour per group on average. The FGI questions were as follows:Introductory prompt 1: Please feel free to tell us what you thought about the DCC classes.Introductory prompt 2: Please tell us about your overall experience and feelings about nursing education using dramatic scenarios.Main prompts:
Please share any feelings you have about any of the dramatic scenes pertaining to topics such as diagnosis, treatment, and prognosis.After exposure to DCC, tell us what you think is the most important competencies that nurses should have.After exposure to DCC, please tell us what you have learned or felt about nursing care for cancer patients.Closing prompt: If DCC is incorporated into nursing education in the future, please tell us what you believe are its advantages and disadvantages.

### 2.5. Data Management and Analysis

The interview was recorded with a digital recorder after obtaining consent from the students, and the researcher transcribed the interview verbatim. Transcripts that could potentially be traced to specific individuals were edited to contain pseudonyms or symbols in place of identifying characteristics.

For qualitative data analysis, the content of the transcripts was read multiple times and analyzed according to the six steps of the thematic analysis method: (1) becoming familiar with the data, (2) generating initial codes, (3) searching for themes, (4) reviewing themes, (5) defining the themes, and (6) naming the themes. Specifically, as the transcripts were read, meaningful phrases and sentences were examined, and important statements were extracted and coded. After deriving an initial code, a high-level concept to which the code could be connected was determined, and the code was classified according to that high-level concept. The conceptual classifications derived as a result of this process were compared and analyzed in order to derive a central theme. The concepts and categories discovered during analysis were collected to extract central themes, and the themes were modified according to their intrinsic implications [19].

### 2.6. Rigor of the Qualitative Inquiry

In order to ensure the rigor of this qualitative research study, credibility, fittingness, auditability, and confirmability were used as evaluation criteria. The credibility of this study was verified by checking with the participants whether the summarized transcripts matched what they said. Fittingness was confirmed based on the researchers’ past clinical experience and research team members’ experience in the clinical field. Auditability was confirmed through regular data analysis meetings between research team members, during which all procedures for data collection and analysis were recorded and reviewed. To ensure confirmability, a research team member checked the recorded files against the transcripts [21].

### 2.7. Ethical Considerations

This study was approved by the institutional review board of the university where the authors were affiliated before conducting the research (SNU-IRB No. 1509/001-009). After all participants were informed of the purpose and necessity of the study, they voluntarily decided whether to participate. Subjects were asked to complete a written informed consent before participating in the study and then were asked to participate in an FGI. Participants were notified that all data would be used for research purposes only and that the data for analysis would be collected anonymously.

## 3. Results

After 51 nursing students took part in the DCC program, their experiences were collected and analyzed through FGIs, and a total of 3 themes and 10 codes were derived (Table 2). After participating in the DCC classes, participants shared their thoughts on care for cancer patients, the role and purpose of nursing, the gap between nurses who provide emotional support and those in reality, and their self-image as future nurses. The experiences of nursing students who attended the DCC classes were classified into three themes: ‘understanding the lives of patients with severe diseases and their families,’ ‘seeing a nursing role model provide patient-centered care,’ and ‘projecting an image of oneself as a future nurse.’

### 3.1. Theme 1: Understanding the Lives of Patients with Severe Diseases and Their Families

The students said that, by watching the emotions, pain, and anguish experienced by patients and their families in real time in the context of the DCC classes, they were able to understand the situation from the patient’s point of view rather than from the nurse’s point of view. The students recalled that, during clinical practice, they mainly focused on nurses’ work through observation and shadowing. While watching dramatic depictions of nursing, however, the experiences of patients and their families were highlighted.

The students said that, by seeing a dramatic depiction of what families go through during the entire process of a patient’s illness, they came to understand the emotional side rather than the rational side of what the families experience. Lastly, the students empathized with the patient and family portrayed in the dramatic scenario, reflected on their negative reactions to past patients they had seen, and were able to sympathize with the patient’s situation.

#### 3.1.1. Understanding a Patient’s Disease Experience

The students said that they were able to understand the course of the disease process through the dramatic scenarios. Seeing the entire disease process helped them develop a deeper understanding of patients and the clinical course of the disease.
*“In the simulation education, I only observed fragmentary nursing techniques, but the drama-combined education was helpful for seeing the nursing situation as a whole.”*(FGI #5, Participant #3)
*“In the ward where I practiced, I had no exposure to surgery-related care because of time conflicts. The training schedule was short, and it was not possible for everyone to see the same things in each ward.”*(FGI #3, Participant #4)


#### 3.1.2. Perceiving Patients as Unique Human Beings

While watching the dramatic depiction of the disease process of cancer patients, nursing students were able to interpret and understand the meaning of the situation calmly from the patient’s point of view rather than from a nurse’s work-oriented perspective. In addition, students said that patients, whom they typically remember by a disease name, type of surgery, or registration number, newly appeared to them as unique human beings who built their own worlds and led their lives in their own ways.
*“Patients also lived as ordinary people leading a good social life, but it is shocking to see that in the hospital, patients are only viewed as X disease, Y surgery, and Z technique.”*(FGI #3, Participant #2)

#### 3.1.3. Having a Sharper Focus on Patients’ Families

The students said that it was through DCC that family members, who were the caregivers for the fictional cancer patients, caught their attention. Caregivers, whom the students had seen occasionally in their short clinical practice experiences, were invisible or hidden behind patients, and the students did not usually get the opportunity to observe them closely. However, the students said that through DCC, they learned about caregivers’ sorrow, sadness, and frustration. The students also said that they realized that caregivers should also be the clients of nursing care and needed emotional support.


*“While watching the drama, I realized that caregivers must have been in pain as well. The patients were physically ill, and I could imagine that the caregivers who were caring for the patients alongside them must have been in distress, and so I thought that nursing care is also needed for caregivers.”*
(FGI #1, Participant #6)

#### 3.1.4. Understanding Patients’ Negative Attitudes

The students said that the occasional hurt of being ignored by patients or caregivers during clinical practice disappeared after seeing patients and caregivers struggling with the realities of a cancer diagnosis. Thus, they were able to sympathize more with patients. The students also said that they once again understood that patients and caregivers are in pain and in a difficult situation. As such, they were able to expand their perspective and accept the negative attitudes of patients and caregivers as part of the disease-fighting process.


*“I remembered one patient who I was in charge of in the past. When a head nurse said that I would be explaining X, the patient said that he was not happy about it. At that time, I thought, ‘My patient is unkind.’ But as I watched the drama today, I could understand him by thinking ‘Ah, how difficult it must have been for him.’ ‘He must have been in a bad mood.’”*
(FGI #3 Participant #5)

#### 3.1.5. Empathizing with the Suffering of Patients and Families

Watching the pain and anguish of patients and their families depicted in the dramatic scenarios, the students said that they were able to empathize with the patient and family members. Some students recalled previous patients whom they cared for during their clinical practice, while other students expressed intense emotions when recalling the experiences of close acquaintances.


*“Watching what families have experienced helped me understand what they were actually going through emotionally rather than rationally.”*
(FGI #1, Participant #3)

### 3.2. Theme 2: Seeing a Nursing Role Model Provide Patient-Centered Care

By seeing a dramatic depiction of nursing, the students were able to recognize the importance of emotional support and therapeutic communication. The nurses in the dramatic scenarios served as role models for students, providing appropriate physical care according to the situational conditions of the fictional breast cancer and lung cancer patients and providing the fictional patients with emotional support. The students said that they learned practical communication skills in a natural way from watching the conversations over the course of the treatment process between the patient and the nurse.

#### 3.2.1. Recognizing the Importance of Emotional Support Using Therapeutic Communication Skills (Empathy, Support, Being Together)

By watching dramatic depictions of emotional support, which students had only vaguely understood through their textbooks, the students acquired practical skills to approach and talk to patients and were able to understand the significance of emotional support. The students said that they learned that speaking to patients with sincere interest and affection could serve as a form of emotional support.


*“By actually observing the nurses treating patients and families with emotional support, I could better understand therapeutic communication that could not be learned through textbooks, and I could wholeheartedly feel the importance of effective communication.”*
(FGI #1, Participant #9)


*“The visual depiction has helped me learn more than I did after just listening to lectures. From these dramatic scenarios, I was able to learn some tips regarding what to do when I talk to patients.”*
(FGI #5, Participant #6)

#### 3.2.2. Modeling the Essence of Nursing through Drama

Through DCC, the students were able to see examples that represented the role of nurses while seeing in real time the practices of professional nurses, who provided holistic nursing, including symptom management and emotional support.

The students said that the DCC classes were not simply lecture-oriented and were an opportunity to naturally acquire communication skills after seeing them applied in practice. They also said that they wished dramatic enactments were developed for other subjects since DCC helped them understand situations that are difficult to grasp from lecture-oriented classes and clinical practice alone.


*“What impressed me was that the nurses showed patients empathy, made eye contact with the patient and family, and held hands, and the nurses themselves were able to control their own emotions, showing a strong shield against getting caught up in the emotions of patients and family members.”*
(FGI #4, Participant #4)


*“When I was receiving simulation-based education, I was busy learning the techniques, so I couldn’t perform emotional care at all, but seeing the actors do it, I felt like I could see things from the holistic perspective, and I think it will be very memorable.”*
(FGI #2, Participants #4)


*“I think the dramatic depictions were helpful because they made me think about what I should do, what I should prioritize in nursing for cancer patients; they were also helpful because they made me think holistically.”*
(FGI #5, Participant #3)

### 3.3. Theme 3: Projecting an Image of Oneself as a Future Nurse

The students said that they felt a discrepancy between the nurses depicted in the dramatic enactments and nurses in reality. During clinical practice, nurses were often busy and did not appear to have the time to empathize with the patients, focusing only on getting the job done.

The students said that, in some cases, there were nurses who treated patients in an unkind manner, providing no emotional support. Some students said that it would be difficult in real-world settings to provide nursing care and speak with patients for extended periods of time, as portrayed in the dramatic enactments, in a very busy hospital environment.

Other students felt that they had an opportunity to reflect on their attitudes in their past clinical practice experiences. The students took the opportunity to reflect on the core elements of nursing through the dramatic situations and to think about their own self-image as nurses by comparing the nurses in reality to those in the dramatic scenarios.

#### 3.3.1. Comparing Nurses in Clinical Practice to Nurses in the Dramatic Scenarios

The students said that nurses were so busy during clinical practice that they rarely saw them provide emotional support to patients. Thus, the fictional nurses depicted in the dramatic scenarios were dissimilar to the nurses that the students had seen in reality. The students also doubted whether the image of nurses as depicted in the dramatic scenarios existed in real life and whether they would be able to embody the fictional nurses’ behaviors.


*“I thought the nurse was really different from the nurses we have actually seen in real life. I participated in a university hospital practicum and actually saw nurses treat patients, but in fact, I don’t think I have seen anyone who cared for patients with the same degree of kindness, sincerity, and empathy as the nurse from the dramatic depiction.”*
(FGI #1, Participant #4)

#### 3.3.2. Reflecting on Oneself in Clinical Practice

The students recalled that they only paid attention to patient records when writing reports, rather than paying attention to patients themselves during clinical practice. With that in mind, they said that DCC was an opportunity for them to reflect on their attitudes in clinical practice. Students also reflected on their attitudes towards the practicum and their views of patients. They regretted that they previously did not view patients as human beings with dignity, instead focusing only on the disease and treatment.


*“We were disappointed seeing ourselves spending too much time sitting down at the EMR and writing notes rather than going to patients to focus on their cases.”*
(FGI #4, Participant #5)

#### 3.3.3. Imagining Oneself as a Future Nurse with Competent Knowledge and Skills

The students said that, because of the dramatic scenarios, they reflected on the images of themselves as future nursing professionals who could provide holistic nursing care. They thought deeply about what nursing truly means, how to provide care, and what kind of nurses they would like to become.


*“I thought that being like the nurse in the drama is what we should learn and become.”*
(FGI #5, Participant #2)

## 4. Discussion

This study was conducted to explore and understand the experiences of nursing students after taking part in DCC, which was an educational course that took a total of 90 min and consisted of two nursing scenarios, one depicting a breast cancer patient and one depicting a lung cancer patient. For each scenario, the course alternated between six lectures and five dramatic scenes.

In the fall of 2015, a total of 67 junior-year students in a nursing college took a total of 6 DCC classes. FGIs were conducted with the 51 students who agreed to participate in the study, and 3 themes were derived using thematic content analysis. The results show that nursing education using drama provided students with deep insights about cancer patients, caregivers, nurses, and care for cancer patients.

The students reported a high level of understanding after they watched dramatic depictions of multiple stages of the disease process experienced by patients and their families combined with short lectures by the professor who conducted the course. The dramatic depictions furnished a safe environment for the students to reflect in-depth upon their experiences and the experiences of others, rather than a stressful and high-pressure environment such as in clinical practice [22].

Through the dramatic depictions, the students witnessed a variety of experiences, including those of patients and caregivers, and felt sympathy and consideration for the fictional characters. As the students empathized with the portrayed patients, they came to understand the behaviors and reactions of past patients they had seen. The dramatic scenarios helped the students to prepare to fulfill their roles as professional nurses, and we believe that the students were able to receive education related to patient care that is not easily communicated in theory-oriented classes. We believe that the students were able to prepare to fulfill their roles as future professional nurses by witnessing dramatic depictions of situations related to diagnosis, treatment, survivor management, and terminal cancer management that are difficult to understand in clinical practice.

The students who participated in the FGIs said that they wanted drama-combined nursing education to be used to raise students’ learning expectations and to be developed for addressing other diseases and situations. In this study, professional stage actors played the roles of patients, families, and nurses in the dramatic depictions, and nursing students watched live performances as audience members. In the future, however, if drama-combined programs are expanded to encompass a variety of other areas and recorded on video so that they can be viewed repeatedly, students will likely have opportunities for deeper reflection and insights [13]. In this study, it was found that students’ understanding and empathy for patients were increased after the DCC class. However, applying only qualitative research analysis is a limitation of this study, and for this purpose, it is recommended to implement a mixed research method that additionally uses a questionnaire tool to quantitatively measure students’ experiences

This study was conducted in collaboration with a professional theater person, and in order to dramatically express the patient’s situation reproduced in the DCC class, it is important to provide accurate education to the theater person on cancer patients. Additionally, for long-term sustainability, collaboration with the theater department in the university is considered to be helpful in conducting the DCC class in the future.

This study was conducted with junior-year students at a nursing college and aimed to help students experience the entire span of the disease process for breast cancer and lung cancer patients. Educational methods using drama are most commonly implemented in the context of specialized pediatric nursing at the beginner level; however, the scope of such methods is expanding, as shown in this study, which applied this method among junior-year undergraduates. This suggests that DCC can even be applied to even more advanced courses for upper-level college students in addition to undergraduates [3,23].

Drama can be incorporated into various educational methods. Compared to existing teaching methods, teaching pedagogy that combines drama and problem-based learning has been proven to result in higher academic achievement and learning satisfaction [24]. Using drama in education can provide opportunities for students to focus on education and stimulate their curiosity. Additionally, the nursing professor performed many roles in the DCC, such as a director, teacher, and researcher, which implied the possibility for developing new educational contents in nursing. The DCC course developed herein constitutes a nursing education method incorporating drama with demonstrated effectiveness. Therefore, it is hoped that nursing education that incorporates drama will be researched and developed for other subjects and students in the future.

## 5. Conclusions

The DCC course developed in this study helped students to understand patients and families more holistically by allowing them to indirectly experience the disease and treatment process through dramatic depictions. By seeing nurses portrayed in drama in the DCC course, students were able to indirectly experience therapeutic communication skills, attitudes, and counseling skills, which prompted them to reflect on their self-image as future nurses. In order to educate students on nursing professionalism and improve their clinical competency, courses using drama should be further developed to provide supplementary education in addition to lectures.

## Figures and Tables

**Figure 1 ijerph-18-09891-f001:**
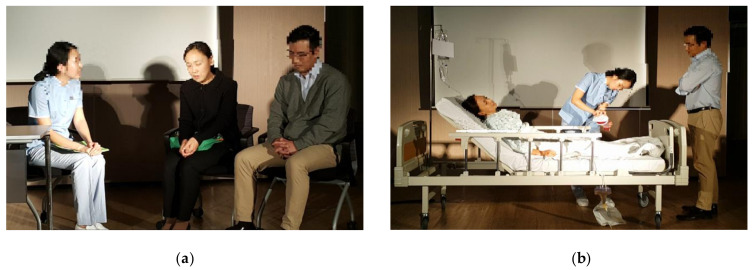
Dramatic scenes from this study: (**a**) Patient and caregiver counseling scene and (**b**) scene of performing nursing intervention after surgery. Reproduced with permission from authors E.E.S. and Y.S.

**Table 1 ijerph-18-09891-t001:** Structure of drama-combined nursing education for cancer care.

Session	Scene	Place	Contents		Time (min)
			Patient	Nurse	
Lecture 1	General information about cancer statistics and etiology				5
Drama scene 1	Cancer diagnosis	Doctor’s office	- Breast cancer diagnosis - Lung cancer diagnosis	- Listening to patient and spouse’s feelings - Information about cancer treatment	10
Lecture 2	The incidence of breast and lung cancer				5
Drama scene 2	Postoperative situation	Nursing ward	- Pain after surgery - Spouse’s concerns	- Postoperative care - Emotional support	10
Lecture 3	Postoperative nursing care (lung care, ambulation, wound care)				5
Drama scene 3	Chemotherapy	Chemo-therapy room	Chemotherapy-induced nausea and vomiting	- Tailored nursing intervention for chemotherapy side effects	10
Lecture 4	Modalities of cancer treatment including operation, chemotherapy, and radiation therapy				5
Drama scene 4	Recurrence	Doctor’s office	Hearing bad news related to cancer treatment	- Therapeutic communication - Provide information	10
Lecture 5	Recurrence rates and palliative care for breast and lung cancer				5
Drama scene 5	Hospice, end-of-life care	Hospice unit	Moments leading to end-of-life	- Therapeutic communication - Supportive care for patient and family	10
Lecture 6	Hospice and end-of-life care				3
Debriefing	Briefly summarized the purpose and contents of the drama-combined nursing educational course on cancer care				10
Closing	Q&A for class, closing				2
6 lectures and 5 dramatic scenes				Total 90 min	

**Table 2 ijerph-18-09891-t002:** Themes and codes from students’ experiences of drama-combined education.

Themes	Codes
Understanding the lives of patients with severe diseases and their families	1. Understanding a patient’s disease experience
	2. Perceiving patients as unique human beings
	3. Having a sharper focus on patients’ families
	4. Understanding patients’ negative attitudes
	5. Empathizing with the suffering of patients and families
Seeing a nursing role model provide patient-centered care	1. Recognizing the importance of emotional support using therapeutic communication skills (empathy, support, being together)
	2. Modeling the essence of nursing through drama
Projecting an image of oneself as a future nurse	1. Comparing nurses in clinical practice to nurses in the dramatic scenarios
	2. Reflecting on oneself in clinical practice
	3. Imagining oneself as a future nurse with competent knowledge and skills

## Data Availability

Data are available upon request.

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
