# Peer review of "The Development and Application of Drama-Combined Nursing Educational Content for Cancer Care"

_ijerph, 2021, doi:10.3390/ijerph18189891_

Round 1
Reviewer 1 Report
Thank you for opportunity to review this manuscript. In general, I consider the study beneficial and the paper well written.
I only have a few comments that could help to strengthen the paper.
1) Could you add more information on conducted intervention? Was DCC provided as a part of compulsory subject? Can the students choose voluntarily if they attend the class? Was any additional CC education provided except described intervention?
2) Could you add more information on data collection? How long after DCC were FGIs conducted?
3) Do you have any recommendation on what needs to be done in the future in terms of possible studies to support you conclusions?
Author Response
- Thank you for your comments.
-
Response to Reviewer 1 Comments
Point 1: Could you add more information on conducted intervention? Was DCC provided as a part of compulsory subject? Can the students choose voluntarily if they attend the class? Was any additional CC education provided except described intervention?
Response 1: DCC class was provided to the students as a compulsory nursing subject. Students who took the practicum of adult health nursing automatically participated in the DCC. However, the participation in the focus group interviews was asked to the students for agreement. I added the additional description about this on page 4, lines 142-146.
Point 2: Could you add more information on data collection? How long after DCC were FGIs conducted?
Response 2: As described in pate 4~5, the DCC was provided six different time points biweekly, the FGIs was followed by each six DCCs. The FGIs lasted 30-40minutes and recorded for analysis.
Point 3: Do you have any recommendation on what needs to be done in the future in terms of possible studies to support you conclusions?
Response 3: Using different kind of research methods such as mixed method needs to be used for future research for supporting our results. I added more description on this on page 11, lines 413-422.
Reviewer 2 Report
Dear Authors,
congratulation for an excellent piece of work! It is fully justifiable that drama-combined nursering education will be needed not only to reduce problems during pandemic restrictions, but has a great potential for future education in health care sciences. You bring an initial evidence and understanding to the usage of arts in nursing education focused on cancer patients. Your study is scientifically sound, clearly written and attractive for readers. It was really a pleasure to read it.
I have only one suggestion concerning the discussion section. It would be appropriate to include at least a short reflection of the strengths / limits of the study, and (concerning the fact it is a qualitative research) also a brief description of the researchers´ roles and experience with the phenomenon researched.
Author Response
Response to Reviewer 1 Comments
Point 1: It would be appropriate to include at least a short reflection of the strengths / limits of the study, and (concerning the fact it is a qualitative research)
Response 1: Thank you for your comments. I added the limitation of this research and implication for the future research on page 11, lines 413-422.
Point 2: also a brief description of the researchers´ roles and experience with the phenomenon researched
Response 2: Thank you so much for your advice. More description on researchers’ role and future research implication on page 11, lines 418-423, and 434-436.
Round 2
Reviewer 1 Report
Thank you for opportunity to review this manuscript. I believe the paper is well written and revised version could be accepted for publishing.